# ENCODING SPATIO-TEMPORAL LOCATIONS WITH ORTHOGONAL FUNCTION REPRESENTATIONS

## ABSTRACT

Complex spatio-temporal dependencies govern many real-world processes – from climate dynamics to disease spread. Modeling these processes continuously using purpose-built neural network architectures, so-called *location encoders*, presents an emerging paradigm in analyzing and interpolating geographic data. In this work, we expand existing spatial location encoders and introduce a new time-informed architecture: the *space-time encoder*. Our method takes in geographic (latitude, longitude) and temporal information simultaneously and learns smooth, continuous functions in space and time. The inputs are first transformed using positional encoding functions and then fed into neural networks that allow the learning of complex functions. We consider, via detailed experimental analysis, (1) how to efficiently combine space and time encodings, (2) the effect of different choices of encoding functions for the time component and (3) frameworks for encouraging orthogonality of feature representations to improve representational power. We highlight the effectiveness and flexibility of the space-time encoder on a range of tasks representing different spatio-temporal dynamics, from climate prediction to animal species classification. Our results show improvements over existing methods for modeling spatio-temporal data, owing both to the space-time encoder architecture and our orthonormal regularizer.

## 1 INTRODUCTION

Geospatial data are central to a wide range of scientific and real-world applications, including conservation of biodiversity (Cole et al., 2023; Dhakal et al., 2025), Earth system modeling (Chen et al., 2024), and agricultural decision-making (Lin et al., 2020). Many tasks in these domains–such as predicting species occurrence patterns, estimating environmental variables from sensor output, or mapping crops–require learning patterns that unfold across space *and time*. Figure 1 exemplifies the importance of modeling spatio-temporal processes on the example of migratory bird species. Tasks like these often rely on sparse, irregular observations anchored to specific spatio-temporal coordinates, from which models must interpolate or extrapolate dense, high-resolution outputs.

With the growing power of deep learning methods and availability of open-access data from remote sensing and citizen science, there has been an abundance of work leveraging deep learning for geospatial tasks (Zhu et al., 2024). As this field has

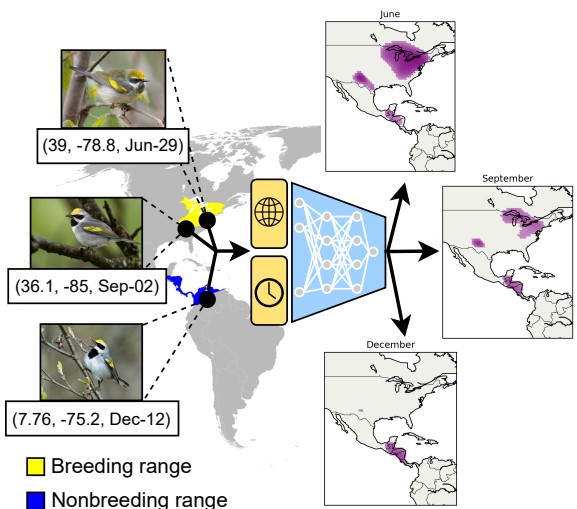

Figure 1: Modeling bird migration patterns, an example for the importance of modeling spatio-temporal processes accurately. This figure shows the learned presence probability map $P(y|\lambda, \phi, t)$ of the Golden-winged Warbler, a migratory warbler species found in the Americas, learned by our *Space-Time Encoder*.

grown, there has been an increasing recognition of the value of designing specialized architectures and methods for encoding the geographical structure of positional information (Mac Aodha et al., 2019a; Mai et al., 2023; Rußwurm et al., 2024; Klemmer et al., 2023) in the form of *location encoders* that can be incorporated into a variety of specific algorithms. However, the adoption of location encoders has so far consisted mainly in incorporating the spatial structure of datapoints situated on the Earth. Geospatial data often contain additional temporal components that affect the observed process, and thus may be expected to change the optimal spatial encoding. Strong temporal dynamics occur in diverse problems such as climate model emulation over time (Watt-Meyer et al., 2023), species distribution modeling of migratory birds (Zuckerberg et al., 2016), and seasonal crop yield forecasting (Cai et al., 2018). Some prior works have leveraged temporal information as input to spatio-temporal encoders (Mac Aodha et al., 2019b; Mai et al., 2020; Dollinger et al., 2025) but to date, there has been no systematic study of different temporal encodings and their integration into location encoder architectures.

In this paper, we construct and evaluate a variety of techniques for integrating time information into location encoders. Motivated by existing studies on the value of orthogonal space-only location encodings (Rußwurm et al., 2024), our work introduces a novel regularizer that encourages representation orthogonality. We run experiments testing the value of encoding spatio-temporal data in the contexts of multi-variate climate model emulation –and as auxiliary information in image classification of animal species. Our contributions are as follows:

- We introduce the *Space-Time Encoder* framework combining location and temporal encodings, including several modular design options.
- We design a novel regularization method that promotes orthogonality between individual components of the representation and can be used with any encoder architecture.
- We show the value of Space-Time Encoders across a variety of datasets featuring geospatial data with a strong temporal component, including ACE (Watt-Meyer et al., 2023), BirdSnap (Berg et al., 2014), and iNaturalist (iNaturalist 2018 competition dataset), and show an improvement on existing baselines. We identify strong time encoding types and find that our orthogonal regularizer can further boost performance.

## 2 RELATED WORK

### 2.1 MACHINE LEARNING FOR EARTH

Recent years have seen a rapid growth in the scale and variety of data describing our planet, ranging from satellite images to the trajectories of cellphone users. Recent work estimates the scale of Earth observation data as around 800 Petabytes in size, growing at a rate of 100 Petabytes per year (Wilkinson et al., 2024). Many diverse types of data have in common the property that they are indexed in space and time. Given such large and diverse data sources, machine learning is becoming an increasingly powerful tool, with promising applications including climate modeling (Bodnar et al., 2025; Lam et al., 2023), agricultural forecasting (Lin et al., 2020; Nedungadi et al., 2025; Kondmann et al., 2021) and land-use change detection (Bai et al., 2023; He et al., 2024). A plethora of methods have been developed for this domain (Rolf et al., 2024), including an increasing number of foundation models aiming to provide general-purpose modeling frameworks for Earth data, such as SatMAE (Cong et al., 2022), ScaleMAE (Reed et al., 2022), Prithvi (Jakubik et al., 2023; Szwarcman et al., 2024), Presto (Tseng et al., 2023), Galileo (Tseng et al., 2025), and AlphaEarth (Brown et al., 2025).

### 2.2 LOCATION ENCODERS

Within machine learning for Earth, a growing body of work has focused on *location encoders*. These are neural network models that take as input geographic coordinates (longitude and latitude), in contrast to e.g. vision models that expect satellite image inputs. Location encoders are lightweight and highly flexible and have shown potential for constructing dense maps from sparse inputs (spatial interpolation) and as "geographic priors" (Mac Aodha et al., 2019a), conditioning an existing model on geographic information (Rußwurm et al., 2024; Mai et al., 2020). Location encoders can also be pretrained on unlabeled Earth data to obtain global "location embeddings" that contain ground conditions captured by the input data (e.g. features recognizable on a satellite image). This can be

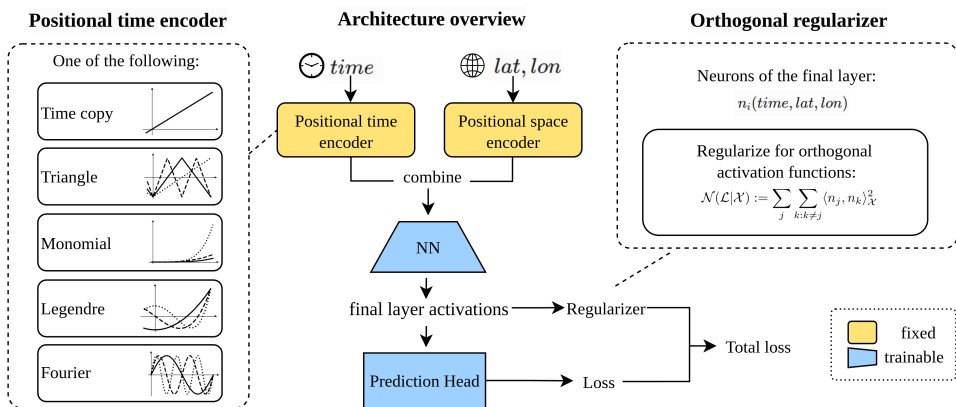

Figure 2: Overview of our proposed *Space-Time Encoder* in the center, with a focus on the orthogonal regularizer (top right panel) and different time encoding configurations (left panel).

facilitated through contrastive learning objectives (Jean et al., 2019; Mai et al., 2023; Klemmer et al., 2023; Vivanco Cepeda et al., 2023) or even untrained "kitchen sink" features (Rolf et al., 2021).

### 2.3 INCORPORATING SPATIO-TEMPORAL STRUCTURE

Spatio-temporal tasks can be broadly split into two categories: (1) Directly predicting some output from space-time coordinates ($y \sim f([lon, lat, time])$) and (2) leveraging space-time coordinates as auxiliary information in a predictive task with primary predictors $x$ ($y \sim f(x, [lon, lat, time])$). For the first category, traditional machine learning approaches include Gaussian Processes (GPs) (Williams & Rasmussen, 2006), Kriging (Matheron, 1969) and other interpolation techniques. More recently, neural networks have been introduced in this setting, promising better scaling and flexibility (Mai et al., 2022; Rußwurm et al., 2024; Chen et al., 2024). For the second category, classical approaches have included domain-specific models that capture the structure of auxiliary information, such as statistical species distribution models (Martínez-Minaya et al., 2018). With more recent deep learning approaches, spatio-temporal metadata has been incorporated as a prior for tasks such as animal and crop type classification from imagery (Mac Aodha et al., 2019b; Mai et al., 2020).

## 3 METHOD

Our methodology addresses the prediction problem of learning functions mapping spatio-temporal coordinates – expressed in longitude, latitude, and time – to an outcome variable of interest, such as air temperature values. In our proposed Space-Time Encoder framework (Figure 2), the spatial and temporal coordinates are mapped to two separate embedding vectors. While the design of the spatial encodings has been the subject of several past studies (Rußwurm et al., 2024; Mai et al., 2020), there is no existing comparison of different temporal encodings for geospatial modeling tasks. We address this research gap in our work.

### 3.1 DEFINITIONS

We here define the property of *orthogonality*, which will be useful in explaining our methodology. Let $\mathcal{F} = \{f_1, \ldots f_N\}$ denote a set of functions mapping a common domain, $D$, to the real numbers, that is, $f_i : D \to \mathbb{R}$. For our purposes, this domain will usually be the sphere $S^2$, the (time) interval $I$, or their Cartesian product $S^2 \times I$. The scalar product of two functions $f, g : D \to \mathbb{R}$ can be defined as the integral of their point-wise product over the domain $D$: $\langle f, g \rangle = \int_D f(x)g(x)dx$. The set of functions $\mathcal{F}$ is an *orthogonal* family if the scalar product of any two distinct elements is zero: $\langle f, g \rangle = 0$, whenever $f \neq g$.

We now consider the neurons of a neural network layer as functions, $n_i : D \to \mathbb{R}$ from the domain of the network to the real numbers. We define a layer to be orthogonal if its neurons represent a set of

| Name | Formula | Orthogonal | Description |
|------|---------|------------|-------------|
| No time | $f_k(t) = 1$ | No | The same constant term in each dimension |
| Time copy | $f_k(t) = t$ | No | The original time coordinate in each dimension |
| Triangular | $f_k(t) = 2 \cdot \left| 2 \left( \frac{kt+1}{2} - \lfloor \frac{kt+1}{2} \rfloor \right) - 1 \right| - 1$ | No | Piecewise linear family of functions |
| Monomial | $f_k(t) = t^k$ | No | Polynomials with a single term |
| Legendre | $f_k(t) = \frac{2k-1}{k} t f_{k-1}(t) - \frac{k-1}{k} f_{k-2}(t)$ | Yes | Orthogonal and normalized version of Monomial |
| Fourier | $f_k(t) = \frac{1}{\sqrt{2}} \sin \left( \frac{\pi kt}{2} \right), \ g_k(t) = \frac{1}{\sqrt{2}} \cos \left( \frac{\pi kt}{2} \right)$ | Yes | Widely used, orthogonal and normalized |

Table 1: Overview of function families for encoding time coordinate grouped into baseline (No time), non-orthogonal time encoders (Time copy, Triangular, Monomial) and orthogonal functions (Legendre and Fourier).

orthogonal functions. Note that an orthogonal layer consisting of a weight matrix $A$, bias vector $b$, and activation function $\sigma$, does not necessarily have an orthogonal weight matrix $A$ and that $A$ being an orthogonal matrix need not imply that the layer is orthogonal.

## 3.2 ENCODING TIME

Our work directly expands on a study using spherical harmonics functions to encode spatial coordinates (Rußwurm et al., 2024). These functions have two desirable properties that we seek to retain in our Space-Time Encoder model: they are both orthogonal and normed - or orthonormal - with respect to the scalar product of square integrable functions on the sphere. Orthonormal functions can represent data with high efficiency due to lack of "overlap" between the information captured.

For the time encoder, we proceed similarly to the space encoder: We choose a family of encoding functions, which we then apply to the time coordinate to obtain an embedding vector. A popular choice for temporal encoding is a basis of Fourier functions, which allows the capture of periodic phenomena such as seasonality. However, based on the dynamics of interest, a different function might be chosen. A neural network can then transform this embedding vector further to obtain the final time embedding vector. The two temporal and spatial embedding vectors are combined and passed through a neural network trained to output the prediction vector.

We compare the effectiveness of several different function families in encoding time, defined formally and compared in Table 1; we summarize them informally here. The *No time* family ignores the time coordinate. The *Time copy* family simply copies the time coordinate across all embedding dimensions. The *Triangular* family spans the set of functions that are piecewise linear in time. The *Monomial* family spans the set of functions that are polynomial in time. Triangular and monomial families are not orthogonal families of functions. The *Legendre* family is a family of polynomials which are orthogonal with respect to the uniform distribution on the interval, and, like the monomial basis, spans the full set of polynomial functions. The *Fourier* family is a family of trigonometric functions that is also orthogonal with respect to the uniform distribution on the interval.

## 3.3 ORTHOGONAL REGULARIZATION

While orthogonal encodings of spatial and temporal inputs help encode information efficiently when passed into a neural network, the orthogonality property is not preserved after passing the encoded coordinates through a neural network.

To encourage the neurons of the last layer $\mathcal{L}$ of a neural network to represent orthogonal functions, we introduce a novel regularizer $\mathcal{N}(\mathcal{L}|\mathcal{X})$. Since we are interested in orthogonality with respect to a data distribution, the regularizer is dependent on the training dataset. This regularizer can be applied to any location encoder architecture, allowing for more flexibility than fixed sets of orthogonal functions such as spherical harmonics functions or Fourier functions which are orthogonal with respect to a fixed probability distribution such as the uniform distribution on the sphere (spherical harmonics) or on the interval (Fourier functions).

The total loss in our experiments is a sum of a supervised loss, $L_{sup}$, such as MSE or Cross-Entropy, and the regularization term weighted by a hyperparameter $\alpha$:

$$L_{total} = L_{sup} + \alpha \mathcal{N}(\mathcal{L}|\mathcal{X}).$$

Just like the total loss, the data-dependent regularizer can be computed on batches of samples.

Given a uniform sample with $N$ elements from the data manifold $\mathcal{X} = \{x_1, \ldots, x_N\}$, one can approximate the integrals that are necessary to test orthogonality: $\langle f, g \rangle = \int_D f(x)g(x)dx \approx \frac{1}{N} \sum_i f(x_i)g(x_i)$. We define a data-dependent scalar product using this Monte Carlo approximation: $\langle f, g \rangle_{\mathcal{X}} := \frac{1}{N} \sum_i f(x_i)g(x_i)$. Using this new scalar product, for a layer $\mathcal{L} = \{n_1, \ldots n_K\}$, we can express an approximate version of the orthogonality condition as: $\langle n_j, n_k \rangle_{\mathcal{X}} - \delta_{j,k} = 0$. By squaring each such equation and adding the equations for all $j, k \in [K]$, we obtain the regularization term:

$$\mathcal{N}(\mathcal{L}|\mathcal{X}) := \sum_j \sum_{k:k \neq j} \langle n_j, n_k \rangle_{\mathcal{X}}^2 \tag{1}$$

The data-dependent scalar product needs $\mathcal{O}(N)$ additions and multiplications. The double sum has $K^2 - K$ terms since we do not sum over terms with $k = j$. Each term consists of a squared data-dependent scalar product. Therefore, evaluating the regularizer requires $\mathcal{O}(NK^2)$ additions and $\mathcal{O}(K^2)$ squaring operations. Since the regularizer is computed in a batched fashion and the width of deep learning layers is much smaller than the number of data-points, this asymptotic complexity bound shows that the regularization term is inexpensive to evaluate.

### 3.4 COMBINING SPATIAL AND TEMPORAL ENCODERS

We now construct orthogonal functions for spatio-temporal coordinates from $S^2 \times I$ using spherical harmonics for the geographic coordinates (spherical domain). There are several examples of orthogonal function sets on the interval (temporal domain) such as Fourier bases and Legendre polynomials–the analogous concept to spherical harmonics on the interval. Orthogonal functions on the sphere and on the interval can be combined to orthogonal functions on the cross product of the sphere and the interval as follows: Consider an orthogonal set of functions $\mathcal{S} = \{f_1, \ldots f_N\}$, which are defined on the sphere, $S^2$, and an orthogonal set of functions which are defined on the interval, $I$, $\mathcal{I} = \{g_1, \ldots, g_M\}$. Then the set $\mathcal{S} \otimes \mathcal{I} = \{f_1 \otimes g_1, \ldots, f_i \otimes g_j, \ldots, f_N \otimes g_M\}$ is orthogonal on the Cartesian product $S^2 \times I$. Here, the tensor product of two functions $f : S^2 \to \mathbb{R}$ and $g : I \to \mathbb{R}$ is defined as $f \otimes g : S^2 \times I \to \mathbb{R}, (x, y) \mapsto f(x) \cdot g(y)$. The proof that $\mathcal{S} \otimes \mathcal{I}$ is indeed orthogonal, follows from Fubini's theorem.

For example, assuming that we chose a spherical harmonics basis, $\{Y_l^m(\theta, \phi)\}_{l,m \in [L] \times [M]}$, for encoding the geographic coordinates and a Fourier basis, $\{\sin(n\pi \cdot t), \cos(n\pi \cdot t)\}_{n \in [N]}$ for the temporal coordinate, the Tensor product of both will have $L \cdot M \cdot N$ elements of the form $Y_l^m(\theta, \phi) \cdot \sin(n \cdot \pi \cdot t)$ and $Y_l^m(\theta, \phi) \cdot \cos(n \cdot \pi \cdot t)$.

## 4 EXPERIMENTS

We assess the performance of the Space-Time Encoder by performing experiments on three datasets that have location and time information: animal image classification with iNaturalist 2018 (iNaturalist 2018 competition dataset) and BirdSnap (Berg et al., 2014), and multi-task climate variable regression with the AI2 Climate Emulator dataset (Watt-Meyer et al., 2023).

### 4.1 CLIMATE VARIABLE REGRESSION

**ACE dataset** Our experiments are based on a subset of the dataset of the AI2 Climate Emulator (Watt-Meyer et al., 2023). The original dataset is composed of 11 climate model simulations each over a period of 10 years. The simulations associate values for 55 climate variables to spatio-temporal location coordinates. The datapoints are arranged on a spatio-temporal grid with a temporal resolution of 6h and spatial grid spacing of 100km. We select 1 year of data from 1 simulation and further select 8 climate variables representing temperatures at different altitudes. We consider the task of interpolation given a sparse set of spatio-temporal coordinates. We used 3% of the available grid points for training set, validation set and test set in equal proportion, i.e. 1% of data, or around 1 million spatio-temporal coordinates randomly sampled from a uniform distribution, for each set.

**Setup** For this task, the training data are mean-centered and scaled by their standard deviation. The models are trained with the Adam optimizer and a Mean Square Error loss function using early stopping with a patience parameter of 5 epochs. We evaluate the models with Root Mean Square Error (RMSE) averaged over the 8 considered temperature variables. Across all datasets, we first consider an encoding baseline in which only the spatial coordinates are used for model prediction but not the time coordinate. We then explore two methods for improving further on this baseline. First, we add a positional time encoding considering five encoding families as shown in 2. Second, we investigate adding an orthogonal regularization term for the last layer during training.

Table 2: Comparison of different time encodings and effect of regularization on ACE. We report mean and standard deviation of average RMSE over 8 temperature variables for 3 runs for regularization weights $\alpha = 0$ (no regularization) and $\alpha = 1$.

| Time enc. | **ACE** $\downarrow$ | |
| | $\alpha = 0$ | $\alpha = 1$ |
| --- | --- | --- |
| No time | 5.75 (0.02) | 5.69 (0.01) |
| Time Copy | 2.61 (0.14) | 2.36 (0.08) |
| Triangular | 3.16 (0.01) | 3.13 (0.01) |
| Monomial | 2.22 (0.01) | 2.01 (0.03) |
| Legendre | **1.63** (0.08) | **1.41** (0.01) |
| Fourier | *1.70* (0.05) | *1.48* (0.02) |
| Reg. better | 6 of 6 | |

### 4.2 SPATIO-TEMPORAL SPECIES DISTRIBUTION MAPPING

**Datasets** The iNaturalist 2018 (iNat2018) dataset (iNaturalist 2018 competition dataset) is a species image classification dataset gathered by citizen scientists, and contains 461,939 observations covering 8,142 categories. The BirdSnap dataset (Berg et al., 2014) contains 49,829 images and species labels for 500 bird species found in North America. For both datasets, we use the same location and time metadata and dataset splits as in Mac Aodha et al. (2019b). Since the iNat2018 test set labels are not publicly available, we split the training set into training and validation sets (in a 90%-10% split) and evaluate on the original iNat2018 validation set. Due to the seasonal pattern of species, similarly to Mac Aodha et al. (2019b), we consider days of the year as the timesteps.

**Setup** For both tasks, we train a space-time encoder on coordinates attached to an image without using the image data. For evaluation, we use two options: For a direct prediction, we evaluated the accuracy of the species distribution classification using only space and time coordinates. For a combined prediction, we multiply the prediction vectors with logits from a pre-trained vision network which we obtain from TorchSpatial (Wu et al., 2024). This corresponds to a combined model $P(y|\mathbf{I}, \lambda, \phi, t) \propto P(y|\lambda, \phi, t)P(y|\mathbf{I})$ where the spatio-temporal species distributions probabilities $P(y|\lambda, \phi, t)$ serve as a prior to an image classifier $P(y|\mathbf{I})$ following Mac Aodha et al. (2019a). The effects of different time encodings and of orthogonal regularization are studied in the same way as for the ACE dataset.

### 4.3 IMPLEMENTATION DETAILS

We use the FCNet architecture which has been proved effective for geospatial interpolation (Mac Aodha et al., 2019b) for the positional encoding networks to encode time and space. For encoding the spatial coordinates we use spherical harmonics functions across all experiments.

**ACE dataset** We use a model with 4 Residual Blocks of size-1024 layers with a total of 8.9 M trainable parameters. We fix the positional space encoding to a set of 400 spherical harmonics ($L = 20$) basis functions also following the approach in Rußwurm et al. (2024). The size of the positional time encoding is fixed to 40 dimensions across all experiments. Both positional encodings are concatenated to form a combined spatio-temporal positional encoding with 440 dimensions.

**Species Distribution Mapping** For the Birdsnap and iNat2018 datasets we use the same general architecture with 4 and 2 Residual Blocks for the respective datasets containing 1024 neurons in each hidden layer. The positional space encoding uses spherical harmonics with 64 dimensions ($L = 8$). The time encoding dimension is fixed to 8 across both datasets and all experiments. For Birdsnap the spatial and temporal encodings are combined using the tensor product approach which yields a combined spatio-temporal embedding of dimension 512. The number of trainable parameters is

Table 3: Effect of different time encodings and of training our models without and with regularization - $\alpha = 0$ and $\alpha = 0.0001$ respectively - on Birdsnap and iNat2018, including baseline results from prior work. We report mean and standard deviation of accuracy over 10 runs. Baseline methods results are taken from Mac Aodha et al. (2019a), in which standard deviations were not reported. For each setup, best models are marked in **bold** and second best models are marked in *italic*.

| Method | BirdSnap ↑ | | iNat2018 ↑ | |
|---|---|---|---|---|
| | $\alpha = 0$ | $\alpha = 0.0001$ | $\alpha = 0$ | $\alpha = 0.0001$ |
| No time | 0.737 (0.003) | 0.738 (0.002) | 0.733 (0.001) | 0.733 (0.001) |
| Time Copy | 0.737 (0.002) | 0.739 (0.004) | **0.742** (0.001) | **0.742** (0.002) |
| Triangular | **0.754** (0.005) | *0.753* (0.002) | 0.736 (0.001) | 0.736 (0.001) |
| Monomial | 0.750 (0.002) | 0.749 (0.004) | **0.742** (0.001) | *0.741* (0.002) |
| Legendre | *0.751* (0.004) | **0.754** (0.004) | *0.740* (0.001) | 0.739 (0.001) |
| Fourier | *0.751* (0.004) | *0.753* (0.004) | *0.740* (0.001) | *0.741* (0.001) |
| **Baseline** Mac Aodha et al. (2019a) | | | | |
| Vision only | 0.701 (-) | | 0.602 (-) | |
| GeoPriors (without time) | 0.717 (-) | | 0.724 (-) | |
| GeoPriors (with time) | 0.718 (-) | | 0.727 (-) | |

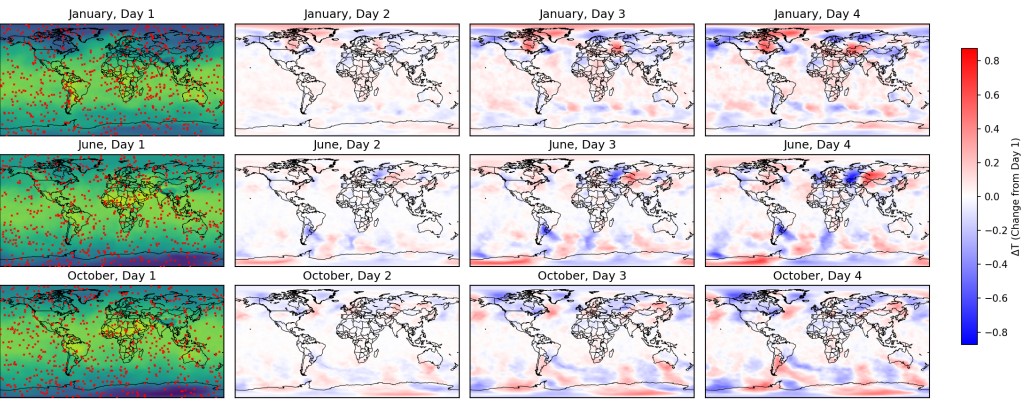

Figure 3: Example results of a *Space-Time Encoder* trained on the ACE dataset. ACE consists of climate model data in a $1° \times 1°$ grid, with 64800 spatial locations. We train a *Space-Time Encoder* to reconstruct a full year of ACE data from $1\%$ of the data per timestep (red dots in the first column). The first colum (Day 1), shows dense reconstruction results on the lowest-to-the-ground temperature layer of ACE, $T_7$. The other columns show changes in subsequent timesteps compared to Day 1, highlighting the capability of our *Space-Time Encoder* to learn smooth and consistent patterns over both space and time.

9.3M. For iNat2018 the spatio-temporal embedding is obtained by concatenating spatial and temporal encoding vectors such that the combined dimension is 72. The model contains 12.9M trainable parameters.

## 4.4 RESULTS & DISCUSSION

Table 2 reports the average RMSE over the 8 temperature variables for the ACE task and Table 3 reports the top-1 species classification accuracies for iNat2018 and Birdsnap over different time encoding methods involving *No time* as baseline, *Time Copy*, *triangular*, *Monomial* as non-orthogonal time encoders, and *Legendre* and *Fourier* as orthogonal time encoders. For each dataset, we also compare results with orthogonality regularization ($\alpha > 0$) and without regularization ($\alpha = 0$). We

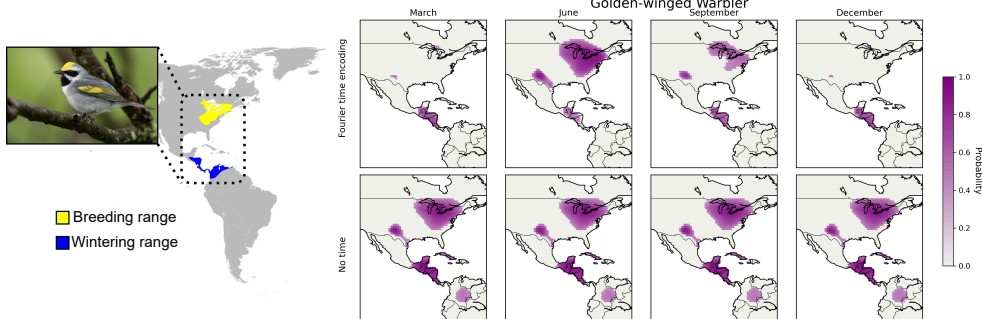

Figure 4: *Left -* Breeding and wintering ranges of the Golden-winged Warbler, a migratory songbird species found in the Americas. (Adapted from "Birds of North America Online": `https://birdsoftheworld.org/bow/home`) *Right -* Occurrence probability maps of Golden-winged Warblers for different months of the year, obtained via our encoder models trained on spatio-temporal coordinates from the BirdSnap dataset. The top row shows probabilities from a Space-Time Encoder using Fourier temporal encodings, the bottom row probabilities from a model not using time information. It is clearly visible that the Fourier Space-Time Encoder learns temporal (migration) patterns of the bird.

further illustrate the ACE reconstruction task and the ability of the *Space-Time Encoder* to learn smooth and continuous patterns in space and time in Figure 3.

We find that **our Space-Time Encoder architecture provides consistent performance improvements over the GeoPrior architecture** (Mac Aodha et al., 2019a) throughout experiments on iNat2018 and Birdsnap. As expected, and confirming results from (Mac Aodha et al., 2019a), adding temporal information improves performance across all dataset. The increases from adding a time component on the species classification tasks are clearly more pronounced than in the baseline results: While adding a time information for the Birdsnap dataset yielded an increase from 71.7% to 71.8% in Mac Aodha et al. (2019a), in our experiments we could observe an increase from 73.7% without time information to 75.4 % with triangular time embeddings. For the ACE dataset, adding the temporal information is essential and improves the average RMSE of the spatial-only encoder by a factor of 3.5 - from 5.75 to 1.63.

**The choice of time encoding can also significantly impact model performance**, with Legendre embeddings reducing average RMSE by half compared to triangular embeddings on the ACE dataset. While the best time encoding differs across datasets, Legendre and Fourier are consistently top or close to top performing across all datasets, which could point to a benefit of orthogonality at the encoding level. Overall, our Space-Time Encoder is the first architecture for geospatial modeling that allows for customizable temporal encodings.

We find that **the benefit of orthogonal regularization can be large, but the effects differ between tasks**. For the species classification tasks on both Birdsnap and iNat2018, the regularizer shows improvements which are not clear cut. However, for the regression task on the ACE dataset, we see a clear benefit of adding orthogonal regularization across all choices of time encodings. This indicates that the regularizer currently performs well for training with an MSE loss on regression tasks but that one has to adapt it for training with a cross-entropy loss on classification tasks.

Table 4: Accuracy (means and standard deviation over 10 runs) of encoders without time encoding and with Fourier time encoding on subsets of the BirdSnap dataset containing only non-migratory and only migratory birds.

| Time enc. | non-migratory | migratory |
|-----------|---------------|-----------|
| No time   | 0.78 (0.006)  | 0.737 (0.004) |
| Fourier   | 0.773 (0.005) | 0.755 (0.003) |

### 4.4.1 ANALYSIS ON MIGRATORY BIRDS

The BirdSnap image classification task allows us to specifically compare performance of our models for classifying migratory and non-migratory bird species. Intuitively, we would expect that performance on migratory bird species, which e.g. spend different season for breeding or wintering in different parts of the world, should benefit from the integration of temporal information into the modeling process. On the other hand, we should be able to model species that reside in the same areas throughout the year sufficiently without temporal information. To test this hypothesis, we compare the performance two of our encoder models, one without time encoding and one with Fourier time encoding, on subsets of the BirdSnap dataset that represent migratory and non-migratory birds respectively. The results, presented in Table 4 and Figure 4, show that adding a temporal encoding does not improve performance for non-migratory bird species, however it improves performance for migratory species by around 1.5 percentage points, thus confirming our hypothesis.

### 4.4.2 EFFECT OF REGULARIZER WEIGHT $\alpha$

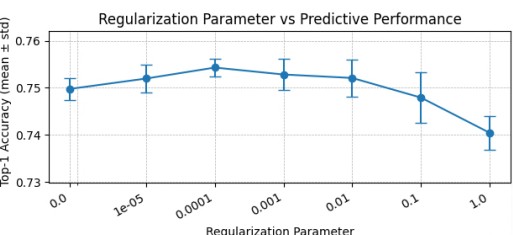

Figure 5: Performance on the BirdSnap dataset for *Space-Time Encoders* trained with different regularizer weight values $\alpha$. Results are averaged over 10 individually trained models.

Experimentally, we found that choosing a too large regularizer weight $\alpha$ can be detrimental to the performance. To better understand the sensitivity of our proposed regularizer, we investigate the performance of our models for different regularizer weights $\alpha$. Figure 5 highlights the performance of *Space-Time Encoders* trained on coordinates from the BirdSnap dataset with different regularizer weight values $\alpha$ when combined with pre-trained vision logits. Compared to models trained without regularization ($\alpha = 0$), using the regularizer with low $\alpha$ values results in consistent Top-1 accuracy improvements. We also observe a slow but increasingly steep performance drop after increasing $\alpha$ over its optimal value around $\alpha = 0.0001$. This indicates that in practice, the hyperparameter $\alpha$ needs to be tuned to the task at hand.

## 5 CONCLUSION

We introduce the *Space-Time Encoder*, a novel neural network architecture working with spatio-temporal coordinates (longitude, latitude and time) as inputs. Our method is aimed at capturing spatio-temporal dynamics, which are crucial in real-world processes such as climate prediction or animal recognition. Our method is a direct expansion of the existing location-only encoder architecture introduced by Rußwurm et al. (2024) and specifically focuses on two aspects: (1) A modular design of the temporal positional encoding, allowing for different encoding functions, and (2) A regularizer that encourages the learning of orthogonal representations within the neural network.

We run experiments on three datasets for two different tasks. First, we test the *Space-Time Encoder* for reconstructing dense spatio-temporal climate model data from sparse data points. Second, we test it as a space-time prior in image classification tasks of animal species, where space-time coordinates are available as auxiliary information. We experiment with different temporal encoding functions, highlighting the best performing options. We also find that our orthogonal regularizer helps to improve the performance of predictive models. Overall, our method outperforms an existing baseline space-time prior in the image classification tasks.

We see the *Space-Time Encoder* as a prototype of a neural implicit representation model, learning continuous, smooth maps in space and time. The location-only encoder (Rußwurm et al., 2024) model also served as the basis for a self-supervised pretraining procedure with globally available pretraining data (Klemmer et al., 2023). In future work, we aim to extend our approach similarly, pretraining *Space-Time Encoders* using globally available, multi-modal and unlabeled geospatial data.

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

## STATEMENT ON LLM USAGE

LLMs have been used to polish and simplify writing, to check for spelling and grammatical errors, as well as for generating and editing LaTeX syntax and to refactor the code provided in the supplementary materials. They have not otherwise been used in this work.

