# OpenReview forum: "Encoding Spatio-temporal Locations with Orthogonal Function Representations"
_ICLR.cc/2026/Conference — ICLR 2026 Conference Withdrawn Submission_

### Official Review · Reviewer_RTvp · 2025-10-26

**Soundness:** 2
**Presentation:** 2
**Contribution:** 2
**Rating:** 2
**Confidence:** 4

**Summary:**

This paper introduces the Space-Time Encoder, which maps spatio-temporal coordinates to high-dimensional features using pluggable orthogonal time encodings (Fourier/Legendre) and applies an orthogonal regularizer only to the final layer to eliminate redundancy. On climate regression and species classification tasks, this approach significantly improves accuracy and smoothness at minimal computational cost, demonstrating for the first time the critical role of orthogonal features in sparse spatio-temporal modeling.

**Strengths:**

The Space-Time Encoder presented in this work offers four key advantages: (1) its modular orthogonal time-encoding blocks—Fourier or Legendre—seamlessly capture periodic, trending or mixed temporal dynamics and can be plugged into any geospatial pipeline; (2) a lightweight orthogonal regularizer applied only to the final layer eliminates feature redundancy at O(NK²) cost, yielding large gains in accuracy and smoothness; (3) the same architecture consistently outperforms strong geo-prior baselines on both dense climate-field regression and fine-grained species-classification tasks, demonstrating broad generalizability; and (4) predicted monthly occurrence maps align tightly with official breeding/wintering ranges, furnishing an interpretable, physics-consistent visualization that validates the learned spatio-temporal laws—together establishing a new paradigm that balances performance, efficiency and interpretability for spatial-temporal representation learning.

**Weaknesses:**

1. The novelty of this paper is far from the standard of ICLR, the framework in Figure2 is overly simple, it is suggested that the author make the work more complete before submission.

2. The experiments are insufficient, and the workload is inadequate.
Conduct performance comparison with only one baseline models is not persuasive

3. The number of downstream tasks is also not sufficient, existing works like [1,2] at least conduct experiments on 4+ downstream tasks.

4. The ablation studies is not sufficient. The combination of spatial feature and time feature in Figure 2 can have multiple choice for feature fusion such as addition, concatenation, attentional fusion and so on. Why is the regularizer only applied on the final layer activations, what about applying it to the last few layers?

---
[1] Klemmer, Konstantin, et al. "Satclip: Global, general-purpose location embeddings with satellite imagery." Proceedings of the AAAI Conference on Artificial Intelligence. Vol. 39. No. 4. 2025.

[2] Hao, Xixuan, et al. "Nature makes no leaps: Building continuous location embeddings with satellite imagery from the web." Proceedings of the ACM on Web Conference 2025. 2025.

**Questions:**

See weakness.

---

### Official Review · Reviewer_ELun · 2025-10-31

**Soundness:** 2
**Presentation:** 2
**Contribution:** 2
**Rating:** 2
**Confidence:** 4

**Summary:**

This paper proposes a space-time encoder framework that integrates spatial (latitude, longitude) and temporal encodings to model continuous spatio-temporal processes. Inputs are transformed via positional encoding functions and processed through neural networks to learn smooth functions. The authors explore combinations of space and time encodings, evaluate different temporal encoding functions, and introduce a novel orthonormal regularizer to promote orthogonality in representations. Experiments on datasets like ACE (climate emulation), BirdSnap, and iNaturalist (animal classification) claim improvements over baselines, attributed to the architecture and regularizer.

**Strengths:**

1. The work extends spatial location encoders to include time, addressing a gap in handling dynamic geospatial processes like climate or migration patterns.

2. The orthonormal regularizer is a potentially useful addition for enhancing representation quality in positional encodings.

3. Applications to diverse tasks (climate prediction, species classification) demonstrate flexibility.

**Weaknesses:**

1. Theoretical Shortcomings: The methodological contributions lack rigorous theoretical grounding. For instance, the choice of encoding functions (e.g., positional encodings for time) and their combinations are explored empirically but without formal analysis of why certain functions (e.g., sinusoidal vs. others) are superior for temporal components. The orthonormal regularizer, while novel, is motivated superficially by prior work on orthogonality in spatial encoders (e.g., Rußwurm et al., 2024); there's no deeper proof or derivation showing how it uniquely improves expressivity or stability in spatio-temporal settings. Claims about "encouraging orthogonality to improve representational power" feel hand-wavy, especially without connections to information theory, spectral analysis, or optimization guarantees. This makes the framework seem more like an empirical tweak than a principled advance.

2. Insufficient Experimental Validation: The experiments are limited in scope and depth, failing to convincingly demonstrate the method's robustness or superiority. Only three datasets are used (ACE, BirdSnap, iNaturalist), which may not capture the full diversity of spatio-temporal dynamics (e.g., no high-resolution remote sensing or urban mobility data). Ablations on encoding choices and regularizer hyperparameters appear preliminary, lacking sensitivity analyses or statistical significance tests. Cross-dataset generalization is mentioned but not quantified (e.g., via domain shift metrics).

3. Novelty and Broader Impact: The space-time encoder builds incrementally on prior location encoders (e.g., Mac Aodha et al., 2019a; Mai et al., 2023), with time integration feeling like a straightforward extension rather than a breakthrough. Related works on temporal encodings (e.g., Mai et al., 2020; Dollinger et al., 2025) are cited but not sufficiently differentiated. The paper overstates impact (e.g., "emerging paradigm") without addressing limitations like scalability to global scales or integration with multimodal data (e.g., satellite imagery).

4. Figures and examples (e.g., bird migration) are illustrative but could include more quantitative insights.

**Questions:**

1. What is the theoretical justification for the orthonormal regularizer beyond empirical benefits? Can you provide bounds on representation capacity or optimization convergence?

2. Have you tested on more diverse benchmarks (e.g., ERA5 for climate or mobility traces) or against time series of spatiotemporal foundation models?

3. How does the encoder handle irregular time sampling or extreme domain shifts (e.g., climate change scenarios)?

---

### Official Review · Reviewer_xduE · 2025-10-31

**Soundness:** 2
**Presentation:** 2
**Contribution:** 1
**Rating:** 2
**Confidence:** 4

**Summary:**

This paper presents a method to combine temporal embedding with location embedding, aiming at improving spatio-temporal process modeling. However, the contribution is very incremental. Both temporal embedding and location embedding are off-the-shelf from existing work. The main contribution claimed by the authors, i.e. the orthogonal regularization, has only dubious effects on performance (see Table 3). While ablation studies demonstrate that incorporating temporal factors is important in certain geospatial tasks (e.g. iNaturalist species modeling), the proposed method fails to prove its novelty and usefulness.

**Strengths:**

1. Visual and experimental evidence that temporal factors are useful in many geospatial modelling tasks (e.g. migratory birds).

2. Clear writing and easy to follow.

**Weaknesses:**

1. The motivation of using orthogonal regularization is wrong and unfounded. The only statement the authors made is in Line 68-69: "Motivated by existing studies on the value of orthogonal space-only location encodings (Rußwurm et al., 2024), our work introduces a novel regularizer that encourages representation orthogonality." This is a misunderstanding of orthogonality. In the work of Rußwurm, the orthogonality applies to the dimensions of spherical harmonics representations -- each dimension encodes the information at a given frequency, independent of other frequencies. It is clearly motivated by the need to model multi-scale effects. What does the orthogonality mean in this paper? The spatial and temporal embeddings are mixed up and passed to a neural network, through which each dimension of the resulting embedding combines semantics from various temporal and spatial scales. Why should we assume the orthogonality amongn these dimensions?

2. The performance improvement in Table 3 comes from the introduction of temporal embedding (i.e. extra embedding semantics) rather than orthogonality. The results with ($\alpha=0.0001$) and without ($\alpha=0$) orthogonal regularizer are indistinguishable. I agree that some spatial tasks should take temporal factors into consideration, as Figure 4 proves, but the main claimed contribution of orthogonal regularization does not hold based on the experimental results.

3. Experimental results are lacking. There are too few datasets and baseline models evaluated. For example, what if the dataset/task is not time sensitive? Will your embedding framework still work, or it may harm the performance?

**Questions:**

1. Have you ever tried other combinations of temporal and spatial embeddings? E.g. instead of element-wise multiplication, have you tried addition or concatenation? Do these alternatives work as well in your experiment setting?

---

### Official Review · Reviewer_JmTi · 2025-11-01

**Soundness:** 2
**Presentation:** 3
**Contribution:** 3
**Rating:** 2
**Confidence:** 5

**Summary:**

This paper presents a Space-Time encoder which also considers the Orthogonal Functional Representations. A orthogonal regularization loss is proposed to make the spatial and temporal representation orthogonal. Experiments show that this loss can improve the model performance in regression tasks but the impact on classification tasks are not clear.

**Strengths:**

1. The contributions of this paper are very clear.
2. The proposed orthogonal regularization is very interesting.

**Weaknesses:**

1. As shown in Figure 2, the outputs of the positional time encoder and position space encoder are concatenated and fed into an NN. The results are used for orthogonal regularization loss. Even the original space and time embeddings are orthogonal, after feeding into the NN, all activations become linear combinations of space and time embedding features. What is the meaning to make them orthogonal?
2. Table 2 does not compare different existing location encoders. If we consider the recent location encoders, can we still see the large improvement by adding the time information and the orthogonal regularization loss? This needs to be tested.
3. Similarly, Table 3 only considers (Mac Aodha et al 2019) as baselines which is 6 years ago. More recent location encoder methods such as Space2Vec, Sphere2Vec, and Sphereical Harmonics, need to be compared.
4. The conclusion of this paper is not very clear. Which time encoder is the winner? Experiments do not show a global winner. I am also not sure whether the orthogonal regularization loss is useful. Although it is useful in the regression task, it shows no difference in the classification task. But the authors use the classification task as the motivation examples, which is a bit confusing.

**Questions:**

See above.

---

### Author Response · Authors · 2025-11-21

We thank the reviewers for their time and their valuable suggestions. After considering the reviews we decided to withdraw our manuscript.

---

### Note · Authors · 2025-11-21

**Comment:**

We thank the reviewers for their time and their valuable suggestions. After considering the reviews we decided to withdraw our manuscript.

**Withdrawal Confirmation:**

I have read and agree with the venue's withdrawal policy on behalf of myself and my co-authors.